# Characterization of the Pro-Inflammatory and Pruritogenic Transcriptome in Skin Lesions of the Experimental Canine Atopic Acute IgE-Mediated Late Phase Reactions Model and Correlation to Acute Skin Lesions of Human Atopic Dermatitis

**DOI:** 10.3390/vetsci11030109

**Published:** 2024-03-01

**Authors:** Amanda Blubaugh, Kathleen Hoover, Sujung Jun Kim, Jonathan E. Fogle, Fatoumata B. Sow, Frane Banovic

**Affiliations:** 1College of Veterinary Medicine, University of Georgia, Athens, GA 30602, USA; alblub@uga.edu (A.B.); khoover@uga.edu (K.H.); 2Boehringer Ingelheim Animal Health, Athens, GA 30601, USA; sujung.kim@boehringer-ingelheim.com (S.J.K.); jonathan.fogle@boehringer-ingelheim.com (J.E.F.); fatoumata.sow@boehringer-ingelheim.com (F.B.S.)

**Keywords:** canine, human, atopic, RNA sequencing, immunoglobulin-E, late phase reactions

## Abstract

**Simple Summary:**

Experimental research using atopic dermatitis (AD) models is required to develop and advance novel therapeutics in AD. Intradermal (i.d.) injections of anti-immunoglobulin E (IgE) antibodies in healthy dogs have been utilized as a model of AD; however, the activated inflammatory and pruritic pathways in IgE-induced skin lesions have not been characterized. This study aimed to characterize the inflammatory transcriptome of experimental acute canine IgE-induced lesions using RNA sequencing and to determine how these correlate to the transcriptome of naturally occurring human and canine acute atopic dermatitis. Acute IgE-mediated lesions had a significant upregulation of pro-, T helper-(Th)1 and Th2 genes and Th2 chemokines. Pathway analysis revealed strong significant upregulation of Janus kinase/signal transducers and activators of transcription (JAK-STAT), histamine, IL-4 and IL13 signaling. Correlation analysis to acute human AD lesions showed a significant moderate positive correlation for anti-canine-IgE 6-h samples (r = 0.53) and 24-h samples (r = 0.47). In summary, acute canine IgE-mediated skin lesions exhibit a multipolar immunological axis upregulation (Th1, Th2 and Th17) in healthy dogs, resembling acute spontaneous human AD lesions.

**Abstract:**

Intradermal injection of anti-immunoglobulin E (IgE) antibodies in dogs grossly and histologically resemble naturally occurring atopic dermatitis (AD). However, the activated inflammatory and pruritic pathways have not been characterized. The objectives of this study were to characterize the inflammatory transcriptome of experimental acute canine IgE-induced lesions and to determine how these correlate to the transcriptome of naturally occurring human and canine acute atopic dermatitis. Biopsies were collected at 6 and 24 h after intradermal injections of anticanine-IgE antibodies to eight healthy male castrated Beagles; healthy and saline-injected skin served as controls. We extracted total RNA from skin biopsies and analyzed transcriptome using RNA-sequencing. Gene expressions of IgE-induced biopsies were compared to that of controls from the same subject (1.5-fold change, *p*-adjusted value ≤ 0.05). Acute IgE-mediated lesions had a significant upregulation of pro-inflammatory (e.g., *LTB*, *IL-1B*, *PTX3*, *CCL2*, *IL6*, *IL8*, *IL18*), T helper-(Th)1/IFNγ signal (e.g., *STAT-1*, *OASL*, *MX-1*, *CXCL10*, *IL-12A*) and Th2 (e.g., *IL4R*, *IL5*, *IL13*, *IL33* and *POSTN*) genes, as well as Th2 chemokines (*CCL17*, *CCL24*). Pathway analysis revealed strong significant upregulation of JAK-STAT, histamine, IL-4 and IL13 signaling. Spearman correlation coefficient for the shared DEGs between canine anti-canine-IgE and human AD samples revealed a significant moderate positive correlation for anti-canine-IgE 6-h samples (r = 0.53) and 24-h samples (r = 0.47). In conclusion, acute canine IgE-mediated skin lesions exhibit a multipolar immunological axis upregulation (Th1, Th2 and Th17) in healthy dogs, resembling acute spontaneous human AD lesions.

## 1. Introduction

Atopic dermatitis (AD) is a pruritic, recurrent and chronic inflammatory skin disease, frequently associated with elevated systemic immunoglobulin-E (IgE) levels, that spontaneously develops in humans and dogs [1]. In both species, the pathogenesis of AD is thought to involve a complex interaction of genetic, immune and environmental factors leading to immune dysregulation and skin barrier dysfunction [1]. 

Immunoglobulin-E plays a role in allergen-induced inflammatory processes in atopic subjects via preformed allergen-specific IgE molecules bounded to the high-affinity IgE receptors (FcεRI) on the surfaces of various immune cells [2,3]. By binding with incoming allergens, IgE acts as an effector for activation of immune cells, chemical mediator release and cytokine/chemokine production. 

Intradermal injection of anti-canine-IgE in healthy dogs results in immediate and late-phase reactions (LPR); LPRs follow 3 to 48 h after allergen challenge and are accompanied by inflammatory cell infiltration that histologically resembles changes seen in naturally occurring canine atopic dermatitis [4,5]. Therefore, cutaneous IgE-mediated LPRs have been used as a screening atopic dermatitis model for studying the anti-inflammatory effect of anti-allergic drugs before entering clinical trials [6,7,8,9]. Interestingly, a pilot quantitative reverse-transcription PCR analysis of canine IgE-mediated LPRs with only a few genes investigated revealed increased expression of proallergic cytokine interleukin-13 (*IL-13*) and CC chemokine ligand 5 (*CCL5*) and *CCL17* [5]. However, to the best of the authors’ knowledge, there have been no evaluations of activated inflammatory and pruritic molecular pathways in IgE-mediated cutaneous LPRs of healthy dogs [10]. 

Given that emerging novel AD treatments are designed to target specific inflammatory mediators or pruritogens, it is imperative to understand the molecular signature of the pre-clinical AD models in humans and dogs. In the past decade, the molecular signatures of numerous human inflammatory skin diseases have been evaluated using gene expression microarrays, which contain a limited number of genes for analysis. Currently, RNA-sequencing (RNA-seq) is commonly used for the evaluation of molecular signatures in skin diseases as it provides the analysis of whole transcriptome and avoids technical issues with microarray (e.g., probe performance). 

In this study, we characterized the activation of early immunologic and pruritogenic pathways in an experimental canine acute IgE-mediated LPR model using RNA-seq of skin biopsy samples sequentially obtained after anti-IgE intradermal injections. Furthermore, we searched through the published RNA-seq databases for acute spontaneous human and canine AD skin biopsies datasets with the goal of performing a comparative analysis of the differentially expressed genes (DEGs) and transcriptional pathways between the acute canine IgE-mediated LPR model and the acute spontaneous AD data.

## 2. Materials and Methods

### 2.1. Patient Inclusion

Eight clinically healthy male castrated research beagle dogs (age 2–3 years) with no previous history of pruritus or skin disease were included in this study. The dogs were housed in the laboratory animal facilities at the university setting under conditions compliant with laboratory animal requirements. All aspects of the study were conducted in accordance with the Institutional Animal Care and Use Committee.

### 2.2. Intradermal Injections and Skin Biopsy Collection of Late Phase Reactions (LPRs)

The IgE-mediated LPRs model was performed as previously described [5] (Appendix A). Dogs were sedated intravenously using medetomidine (Domitor, Pfizer, Exton, PA, USA) and baseline healthy skin biopsies were obtained seven days before the anti-IgE/control intradermal injections. Two intradermal injections of 0.05 mL of anti-canine-IgE polyclonal antibodies (0.08 mg/mL, goat anti-canine IgE AHP946, Bio-Rad Laboratories, Inc., Hercules, CA, USA) and 0.05 mL phosphate-buffered saline (diluent, negative control; Sigma-Aldrich, St Louis, MO, USA) were administered on one side of the thorax and blindly evaluated by an investigator (XX); histamine (0.1 mg/mL, Sigma-Aldrich, St. Louis, MO, USA) served as a positive control. 

The order of injections was randomized for every dog using statistical computer software (GraphPad Prism version 8.0, Boston, MA, USA) and the investigator FB) was blinded during intradermal injection evaluations. Clinical scoring by blinded investigator (FB) involved a global wheal score (GWS) and LPRs [5,6]. 

Skin biopsy samples were collected from IgE-mediated cutaneous reactions at 6- and 24-h post-injections; the 6- and 24-h saline sample served as negative controls. All biopsies were bisected; one half was placed in 10% neutral buffered formalin, and the other was immersed immediately in RNALater and kept frozen until RNA extraction.

### 2.3. Histopathology

Five-micrometer paraffin-embedded sections were stained with haematoxylin and eosin for examination of inflammatory cells. All slides were evaluated by a board-certified pathologist and the results were expressed as a number of positively staining cells in the superficial dermis per 10 consecutive 40X high power fields (HPF), excluding endothelial cells and adnexa. 

### 2.4. RNA-Sequencing Analysis 

The sample size for RNA-seq was determined to be sufficient to provide at least 80% power to detect a significant 1.5-fold difference in values (mRNA transcription) between pre- and post-injections skin biopsy samples using ssizeRNA [8]. Total mRNA was extracted using miRNAeasy kit from Qiagen (Qiagen, Valencia, CA, USA) following the manufacturer’s specifications. Only samples with a 260/280 ratio of ~1.8–2.0 (RNA) and showing a ribosomal integrity number (RIN) above 7 were subjected to further library preparation and sequencing. Forty RNA samples were analyzed in this study: eight healthy non-treated skin samples, eight samples from 6 and 24 h post-intradermal saline (control) and eight samples from 6 and 24 h post-intradermal anti-IgE injection. Sequencing was performed on NovaSeq 6000 with 150 paired-end base pairs (32 samples) and with the NextSeq 2000 with 75 paired-end base pairs (8 samples) according to the manufacturer’s protocol (Illumina, San Diego, CA, USA). For RNA-seq data analysis, please see Appendix A.

### 2.5. Quantitative Reverse-Transcription PCR Analysis

To confirm the gene expression results from RNA-seq data in this study, Quantitative real-time polymerase chain reaction (qRT-PCR) was performed for anti-canine IgE skin reactions at 6- and 24-h timepoints for selected genes. Expression of CCL2, C-X-C motif chemokine ligand 10 (CXCL10), CCL17, IL-1B, tissue necrosis factor alpha (TNF-α) and IL-33 was measured by quantitative reverse-transcription PCR (qRT-PCR; see Appendix A).

### 2.6. Correlation Analysis to Previously Published Transcriptome Data Using RNA Sequencing for Acute Skin Lesions in Human and Canine Atopic Dermatitis

To assess the molecular similarity of canine IgE-mediated LPRs and acute lesions of human/canine spontaneous AD skin, Gene Expression Omnibus (GEO) repository (http://www.ncbi.nlm.nih.gov/geo; accessed on 1 November 2023), a public gene-related database was searched for original RNA seq expression data for human and canine acute atopic dermatitis using the following selection criteria: (1) the study should contain expression data of normal skin biopsy and acute AD (e.g., lesion developed in less than 72-h duration, lack of skin lichenification and histopathologic features of chronic AD lesions) lesional skin biopsy without any immunomodulatory drugs provided to patients at the time of biopsy (i.e., to avoid potential effect of drugs); and (2) samples should be from Homo Sapiens and Canis lupus; and (3) the transcriptome expression profiling by RNA-seq ≥ 75 paired end base pairs reads so it can be compared to the sequencing platform performed in this study. 

Correlation between significantly upregulated/downregulated DEGs (FC = +/−1.5; FDR < 0.05) of anti-canine-IgE reactions at 6- and 24-h and spontaneous human and canine AD acute lesional skin specimens was evaluated using Spearman correlation coefficients on log_2_-transformed levels as previously described. Data were presented in scatterplots with estimated linear regression and a 95% confidence interval.

### 2.7. Statistical and Bioinformatics Analyses

Results of the clinical evaluation (GWS and GLS), as well as cell counts for each time point, were compared with using nonparametric repeated measures one-way ANOVA (Friedman test) with a level of significance set at *p* < 0.05. Normalization of RNA-seq data and DE analysis between different conditions (e.g., healthy, control, IgE, atopic lesional) was performed using empirical Bayes linear model, DESeq2, as implemented through the vignette [11]. A false discovery rate (FDR) of less than 0.05 and fold change (FC) of +/−1.5 or greater was used to determine differentially expressed genes (DEGs). Phenotypically unbiased evaluation of gene set variation between groups was performed with Gene Set Variation Analysis (GSVA) [12,13]. Functional enrichment analysis for pathway identification was done using the Metacore platform for all DEGs [14,15]. 

## 3. Results

### 3.1. Global Wheal Score, Late Phase Reaction Scores and Histopathological Examination

Intradermal injections of anti-canine-IgE and histamine resulted in positive wheal and erythema reactions on the thorax in all eight dogs (Figure 1a,c). There were no wheal and flare reactions observed after the intradermal injections of phosphate-buffered saline (control, Figure 1a). Anti-canine-IgE injections induced significant LPRs at 6 (Wilcoxon matched-pairs signed-rank test, *p* = 0.007 for saline, *p* = 0.005 for histamine) and 24 (*p* = 0.044 for saline, *p* = 0.044 for histamine) hours, compared to phosphate-buffered saline and histamine (Figure 1b,d), respectively. A blinded histological evaluation of 6- and 24-h anti-IgE-associated LPRs revealed a significant increase in total leukocyte superficial dermal cell infiltrate (Figure 1e; *p* = 0.041 for 6-h, *p* = 0.003 for 24-h), as well as lymphocyte (Figure 1g; *p* = 0.022 for 6-h, *p* = 0.007 for 24-h), counts compared with corresponding saline timepoints. Elevated eosinophil numbers were observed in 6- and 24-h IgE-mediated LPRs, but a significant increase was only identified in 24-h IgE LPRs (Figure 1f; *p* = 0.011, respectively).

### 3.2. RNA-seq Molecular Profiling of Anti-Canine-IgE- and Saline-Mediated Late Phase Reactions (LPRs)

Using criteria of fold change/FC of +/−1.5 and false-discovery rate/FDR < 0.05 to define differentially expressed genes/DEGs, we identified 5042 (6 h; 2481 up- and 2561-downregulated) and 3551 DEGs (6 h; 1970 up- and 1581-downregulated) in anti-canine IgE groups versus healthy normal skin, respectively. In contrast, saline injections induced lower number of DEGs at 6 (total 1540; 775 up- and 764-downregulated) and 24 (total 1152; 591 up- and 561-downregulated) hours compared to normal skin. A principal component analysis (PCA), as depicted by the fitted ellipses, demonstrates separation within each group and clear deviation of groups between healthy, saline, and IgE at the 6- and 24-h timepoints after batch correction and normalization (Appendix A).

Overall, both injections (anti-canine-IgE and saline) induced significant differences in treated versus healthy skin for numerous immune genes, with anti-canine-IgE-induced lesions generally producing stronger immune responses (Figure 2 and Appendix A). These include significant upregulations of pro-inflammatory (*IL-1β*, *IL-8/CXCL8*, *IL-6*, *IL-18*) and Th1 (*CXCL10*, *STAT1*, *MX1*, *CCL4*) markers.

Anti-canine-IgE reactions showed stronger increase in Th2-related markers (e.g., *IL-13*, *IL-4R*, *IL-5RA*, *CCL5*, *CCL13*, *CCL17*, *CCL24*, *POSTN*, *STAT6*) compared to saline group; saline group did not reach significance for multiple Th2-related cytokines and chemokines (e.g., *IL-13*, *IL-5RA*, *CCL24*, *POSTN*, *STAT6*). In addition, there was an upregulation of *IL-9R* (FC = 7.9) only for anti-canine-IgE-mediated reaction at 24 h; *IL-9* (FC = 4.3) was upregulated as well, however, did not reach statistical significance (FDR = 0.05). Although significant modulation of several Th17/Th22-related markers (e.g., *IL-17RA*, *IL-23A*, *CCL20*, *S100A12*) was seen across both groups, there were no significant changes in key Th17 (i.e., *IL-17A/IL-17F*) and Th22 (i.e., *IL-22*) markers in both anti-canine-IgE and saline at 6 and 24-h groups compared to healthy control.

We next evaluated genes associated with epidermal barrier differentiation and lipid synthesis [16,17] (Appendix A). While both interventions, the anti-canine IgE and saline injections, induced significant changes in epidermal barrier genes compared to healthy, only anti-canine IgE LPRs showed significant downregulation of terminal differentiation (*FLG*, *FLG2*, *CDSN*, *LOR*, *LCE1E*, *LCE6A*, *CAPN1*, *ST14*), gap/tight-junctions (*CLDN1*, *CLDN4*, *CLDN5*, *GJB5*) and lipid metabolism/biosynthesis markers (*ELOVL1*, *ELOVL2*, *ELOVL6*, *ALOXE3*). Saline and anti-IgE injections increased stress-associated (alarmin) keratins (*KRT6A*, *KRT6B*) and *TIMP-1*, a tissue inhibitor of metalloproteinase 1.

In the anti-canine IgE skin biopsy samples, several significantly up-regulated noncytokine pruritogens (Appendix A) were genes encoding nerve growth factor (NGF) and its high affinity receptor *NTRK1* (*TrkA*), the proteases cathepsin S (*CTSS*), chymase (*CMA1*) and the tryptic peptidase mastin, periostin (*POSTN*) and the enzymes involved in leukotriene-B4 (*LTB4*) synthesis (5-lipoxygenase [*ALOX5*] and its activating protein FLAP [*ALOX5AP*]) and the cysteinyl leukotriene receptor 1 (*CYSLTR1*) and 2 (*CYSLTR2*) (Appendix A). Both interventions (anti-canine IgE and saline injections) significantly upregulated the enzyme involved in histamine metabolism, histidine decarboxylase (*HDC*), and histamine receptor 1 (*HRH1*); however, only anti-canine IgE LPRs showed upregulation of histamine N-methyltransferase (*HNMT* at 24 h) and histamine receptor 4 (*HRH4* at 6 h). Interestingly, anti-canine IgE injections significantly downregulated genes encoding substance P (*TAC1*), MAS-related GPR family member X2 (*MRGPRX2*) and endothelin 1 (*EDN1*). No significant differences in IL-31 expression existed between any interventions and healthy normal skin. 

### 3.3. Pathway and Enrichment Analysis of of Anti-Canine-IgE- and Saline-Mediated Late Phase Reactions (LPRs)

To perform pathway-level comparisons across the molecular skin profiles induced by each agent, we conducted a Gene Set Variation Analysis (GSVA) using previously published immune gene-sets for Th1, Th2, Th17 and Th22_IL22 [16,17] (Figure 3). 

Intradermal injection of saline induced elevated Th1 response at 24 h; there was no significant increases in the Th1-, Th2-, Th17- and Th22-related pathways for any other time point. Anti-canine IgE injections showed significant upregulation in Th1-, Th2- and Th17-regulated genes at 6 and 24 h; the Th22_IL-22 pathway was significantly elevated only at 24 h.

To understand whether the DEGs are significantly enriched in the transcriptional enrichment and pathway analysis, upregulated and downregulated DEGs were analyzed by using Metacore software. Skin lesions induced by anti-canine-IgE injections at 6 and 24-h induced 47 and 59 significantly upregulated process networks, respectively (Appendix A). In contrast, intradermal saline injections induced weaker responses at 6 and 24-h with only 24 and 28 significantly upregulated process networks, respectively (Appendix A). 

The top 20 most upregulated process networks (Appendix A) for anti-canine-IgE injections for both time points were related to the immune system, such as chemotaxis, JAK-STAT pathway, lymphocyte proliferation, leukocyte chemotaxis, interferon signaling, phagocytosis, antigen presentation, neutrophil activation, IL-4 signaling, NK cell cytotoxicity, T helper differentiation and innate inflammatory responses. There was also upregulation of inflammatory responses via IL-5-, IL-13- and IgE-signaling, histamine signaling and Th17 cytokine immune response. 

The intradermal injections of saline control upregulated immune responses as well, such as chemotaxis, JAK-STAT pathway, cell-matrix interactions, extracellular remodeling, connective tissue degradation, phagocytosis, antigen presentation, neutrophil activation, innate inflammatory responses, Th17 cytokine immune response and IFN-y signaling. However, the number of upregulated genes in these pathways induced by saline control was significantly lower compared to anti-canine IgE-induced pathways. Furthermore, the robust immune response of interleukin signaling pathways associated with Th2 pathways, such as IL-4-, IL-5-, IL-13- and IgE signaling pathways, that were increased in anti-canine-IgE reactions were not observed in saline control reactions at any time point.

Finally, the downregulated DEGs from anti-canine-IgE reactions at 6 and 24-h induced 11 and 9 significantly downregulated process networks, respectively (Appendix A). The downregulated process networks were associated with neurogenesis synaptogenesis, neurogenesis axonal guidance, synaptic contact, transmission of nerve impulse, potassium and calcium transport, confirming the functional connection between the IgE activation of the immune system and free nerve endings in the skin. In contrast, there were only 2 and 4 significantly downregulated process networks from saline control DEGs at at 6 and 24-h, respectively (Appendix A); these networks were associated with translation initiation, muscle contraction, actin filaments and skeletal muscle development.

### 3.4. Quantitative Real-Time Polymerase Chain Reaction (qRT-PCR)

Quantitative reverse-transcription PCR performed for selected genes (*CCL2*, C-X-C motif chemokine ligand 10 (*CXCL10*), *CCL17*, *IL-1B*, tissue necrosis factor alpha (*TNF-α*) and *IL-33*) at the 6-h and 24-h IgE timepoints were strong, at 0.90 (*p* = 0.01) and 0.93 (*p* = 0.007) correlation coefficients with a 95% confidence interval for Pearson correlations, respectively, demonstrating efficacy of RNA-seq to capture expression data accurately.

### 3.5. Correlation Analysis to Acute Skin Lesions in Human and Canine Spontaneous AD

The search of the database identified only a single spontaneous human AD study that analyzed 38 healthy and 11 acute skin lesional AD samples using RNA seq at 125 base pairs; no spontaneous canine AD RNA seq studies that incorporated healthy and acute lesional AD skin samples using RNA seq analysis were found. Therefore, we investigated the overlap between these human orthologues of canine DEGs in anti-canine-IgE reactions at 6 and 24-h and a previously published transcriptome skin data from acute spontaneous human AD skin lesions using criteria of −1.5 ≥ FC ≥ 1.5 and FDR < 0.05. Spearman correlation coefficient for the shared DEGs between canine anti-canine-IgE and human AD samples revealed a significant moderate positive correlation for anti-canine-IgE 6-h samples (1198 shared DEGs; r = 0.53; *p* < 0.001; Figure 4a) and 24-h samples (1020 shared DEGs; r = 0.47; *p* < 0.001; Figure 4b).

## 4. Discussion

This is the first global molecular profiling study of inflammatory skin lesions activated through IgE signaling in the skin of healthy dogs. Despite similar activation of pro-inflammatory genes in both groups, our data revealed significant variations in cellular infiltration and expression of immune and barrier genes between anti-canine-IgE and saline control skin lesions. With AD representing one the most prevalent inflammatory skin disease in humans and dogs, there is a need for the development of screening experimental models for studying the anti-inflammatory effect of anti-allergic drugs before entering clinical trials, like anti-canine-IgE reactions. 

Historically, AD has been characterized as a biphasic disease with a Th2-to-Th1 transition from acute to chronic AD stage; however, there are conflicting data [19,20]. To elucidate the transition from acute to chronic AD disease stages and the factors and mechanisms that shape chronic inflammatory activity, Tsoi et al., recently performed RNA sequencing on acute and chronic AD lesions within the same individuals [21]. The results by Tsoi et al., in 2020 showed that the changes accompanying the transition from nonlesional to acute to chronic inflammation in AD are quantitative rather than qualitative; approximately 74% of the genes dysregulated in acute lesions remain or are further dysregulated in chronic lesions [21]. All the major Th1, Th2, Th17 and Th22 responses were progressively heightened from nonlesional AD to acute and then chronic AD lesions, whereas nonlesional AD was enriched in Th2 and Th17 responses [21]. In the study of this report, intradermal injections of anti-canine IgE in healthy dogs induced acute multipolar Th polarization in the skin with early upregulation of Th1, Th2 and Th17 pathways at 6 h and additionally at 24 h the Th22_IL-22 pathway. In addition, anti-canine IgE-mediated reactions transcriptomic profile was compared with the spontaneous acute human AD transcriptome to compare how well the IgE-induced skin lesions represent human AD. Previous studies utilizing murine AD-like models revealed that murine transcriptomes represent only 37%, 18%, 17%, and 11% of the human meta-analysis–derived atopic dermatitis profile (MADAD) for IL-23–injected, NC/Nga, oxazolone (OXA)-challenged, and ovalbumin (OVA)-challenged mice, respectively [22]. Interestingly, there were 1198 overlapped DEGs (−1.5 ≥ FC ≥ 1.5 and FDR < 0.05) with moderate positive correlation (r = 0.53) between the transcriptome of anti-canine IgE skin lesions at 6-h and acute human AD in our study. Although IgE-induced lesions did not capture all immune and barrier aspects of chronic spontaneous human AD, the results of this study demonstrate that acute IgE-mediated signaling in skin induces a wide array of inflammatory axes, including Th2 activation, and could be suited in pre-clinical studies to evaluate Th2 AD-centric axis and how it communicates with other activated immune responses in AD patients. 

Histologically, acute AD skin lesions in humans and dogs exhibit spongiosis with mild to moderate acanthosis in addition to a superficial perivascular infiltrate of lymphocytes, dendritic cells and macrophages [23,24,25]. In addition, mast cells can show degranulation, and occasionally eosinophils may be present with rare neutrophils [23,24,25]. Anti-canine IgE acute skin lesions in this study featured similar changes with mild acanthosis and superficial dermis expanded by mild edema, intermixed with neutrophils, eosinophils, lymphocytes, and plasma cells; eosinophils and mononuclear cells dominated the late-phase reactions at 24 h. Eosinophil recruitment to allergic inflammation sites in the skin and other tissues is driven by IL-5 signaling and by eotaxins (CCL11, CCL24, CCL26) and other chemokines such as regulated on activation, normal T cell expressed and secreted (RANTES; CCL5) and CCL3; these chemokines bind to eosinophils via the ß-chemokine receptor CCR3 [26]. The significant eosinophilic inflammation in the IgE-mediated skin lesions in this study compared to controls can be explained by the strong upregulation of IL-5 signaling pathway with *IL-5RA* (FC = 33.5 and FC = 52.4 at 6- and 24-h reactions) as well as chemokines *CCL3*, *CCL11* and chemokine receptor *CCR3* (FC = 113.6 and FC = 79.5 at 6- and 24-h reactions)

As previously mentioned, AD has long been considered a Th2 disease. A recent study revealed that IL13 is the dominant Th2 cytokine in spontaneous acute and chronic human AD skin lesions [18]. Currently, IL-13 is considered the central driver of Th2 inflammation in human and canine atopic skin [18,27]. Interleukin 13, along with receptors IL4R and IL13Rα1, was the dominant Th2 cytokine in the anti-canine IgE skin lesions in this study, whereas IL-4 was not affected by the IgE-mediated activation. Interleukin 31 is another Th2 cytokine with a dominant pruritogenic effect across different species, such as humans, primates, mice, and dogs [28,29]. Increased IL-31 serum levels have been observed in some AD dogs; a subset of dogs with AD had no detectable levels of IL31 in circulation [30]. In this study, the RNA-seq showed no significant upregulation of *IL31* in any groups. Interestingly, previous murine and canine studies have also had issues amplifying IL31 in the skin of atopic dogs using RNA-seq [31,32,33]. It is possible that cutaneous *IL-31* mRNA is temporary, unstable, and unpredictable or that the *IL-31* sequence in the current canine genome utilized in RNA-seq analysis is not accurate and further studies using immunohistochemistry or immunofluorescence to target IL31 protein presence can be performed to elucidate the presence of IL31 in anti-canine IgE activated skin lesions.

Lesional AD skin features skin barrier dysfunction, characterized by multiple factors, including reduction of epidermal barrier proteins, ceramides, adhesion intercellular proteins and antimicrobial peptides [16,17,18]. Th2 cytokines, IL-4 and IL-13, have been shown to downregulate the expression of skin barrier proteins and lipids in human keratinocyte cultures [34]. In contrast to saline control, anti-canine IgE-mediated inflammation significantly downregulated the expression of several epidermal proteins (*FLG*, *FLG2*, *CDSN*, *LOR*), tight-junction adhesion molecules (*CLDN1*, *CLDN4*, *CLDN5*) and lipid metabolism/biosynthesis markers (*ELOVL1*, *ELOVL2*, *ELOVL6*) in this study. The increased IL-13 expression with the lack of upregulation of IL-4 in anti-canine IgE skin lesions is likely the result of the observed skin barrier changes, resembling barrier dysfunction in spontaneous AD lesional skin in dogs and humans.

Some of the limitations in this study included a small sample size and a lack of identification of protein expression. We determined the sample size using the sample size calculator recommended for RNA-seq studies. In addition, the current canine protein multiplex assays have a limited number of markers available to analyze, in contrast to human assays that allow evaluations of up to 300 proteins [35]. Furthermore, the batch effects present between samples in this study (specifically 75PE and 150PE reads depending on the sample) could cause a limit in the proper detection of DEGs by confounding the analysis. However, utilizing ComBat-seq, an extension of the ComBat approach, is optimal in this scenario as it works well on smaller sample sizes and is robust against outliers [36].

## 5. Conclusions

In summary, the molecular characterization of experimental models utilized to evaluate therapeutics for spontaneous inflammatory skin diseases is essential for future mechanistic and immunomodulatory pre-clinical studies in humans and dogs. In this study, IgE-mediated skin lesions appear to mediate similar T-helper pathways and barrier changes to that of acute spontaneous human AD lesions through examination of the RNA-seq transcriptome; IL-13 represents the dominant Th2 cytokine in this model. Although the initial pilot study using qRT-PCR revealed that prednisolone reduced IL-13, IL-5, CCL2, CCL5 and CCL17 in IgE-mediated skin lesions, further molecular studies using RNA-seq should investigate the effect of established canine anti-allergic drugs (e.g., glucocorticoids, oclacitinib as JAK inhibitor) to validate the canine IgE model. 

## Figures and Tables

**Figure 1 vetsci-11-00109-f001:**
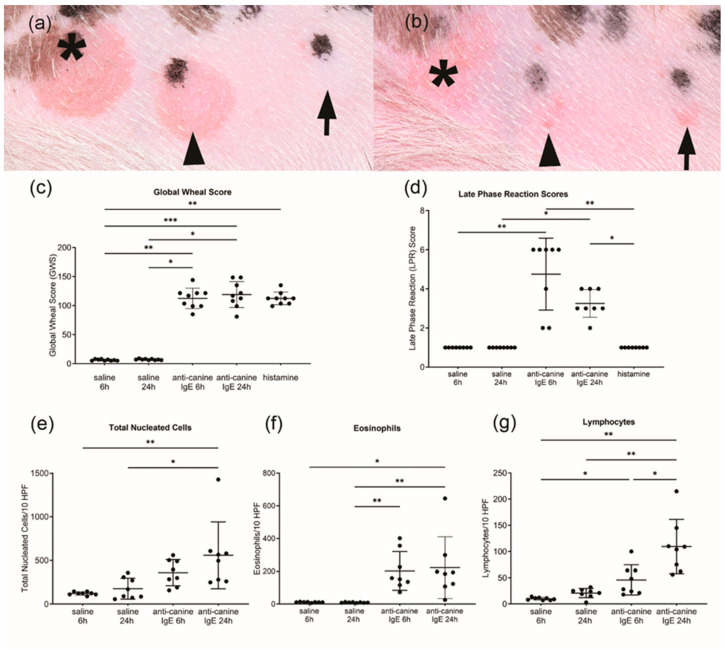
Clinical images of wheal and flare responses after 20 min GWS; (**a**) and 6 h LPRs, (**b**) of injected compounds anti-canine IgE (asterisk), histamine (arrowhead) and phosphate-buffered saline (arrow). Anti-canine-IgE injections induced strong global wheal scores GWS; (**c**) and late phase reactions LPRs; (**d**) at 6- and 24-h, compared to phosphate-buffered saline and histamine control. (**e**–**g**) Histopathological evaluation of the number of inflammatory infiltration cells (**e**), eosinophils (**f**) and lymphocytes (**g**) in saline and anti-IgE late phase reactions (LPRs). * *p*-adj < 0.05, ** *p*-adj < 0.01, *** *p*-adj < 0.005.

**Figure 2 vetsci-11-00109-f002:**
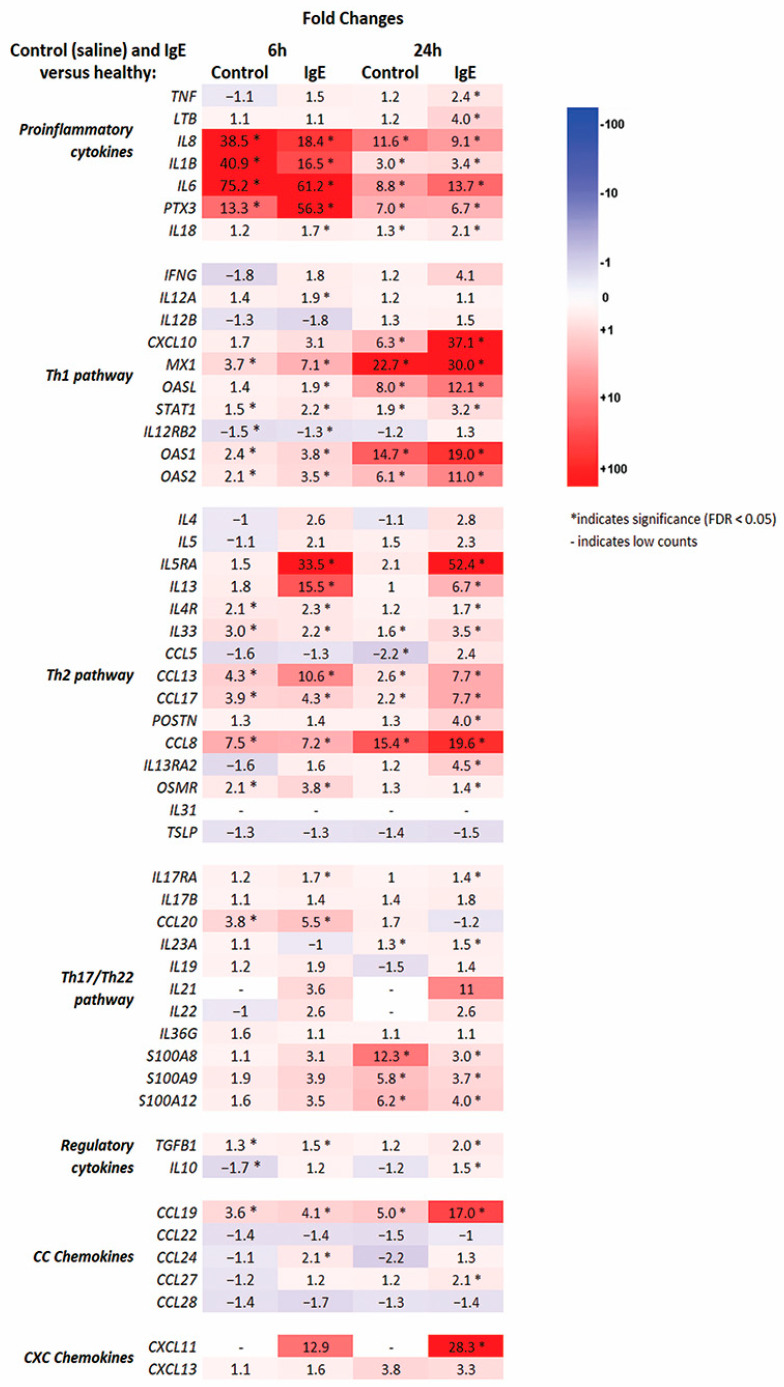
Expression of selected relevant cytokine, chemokine and receptor genes in anti-canine-IgE and saline cutaneous reactions at 6 and 24 h after intradermal injections. Genes are arranged by their dominant function or family and color responds to downregulation (dark blue) and upregulation (bright red); fold changes (FC) with asterisks are statistically significant at * false rate discovery (FDR) < 0.05.

**Figure 3 vetsci-11-00109-f003:**
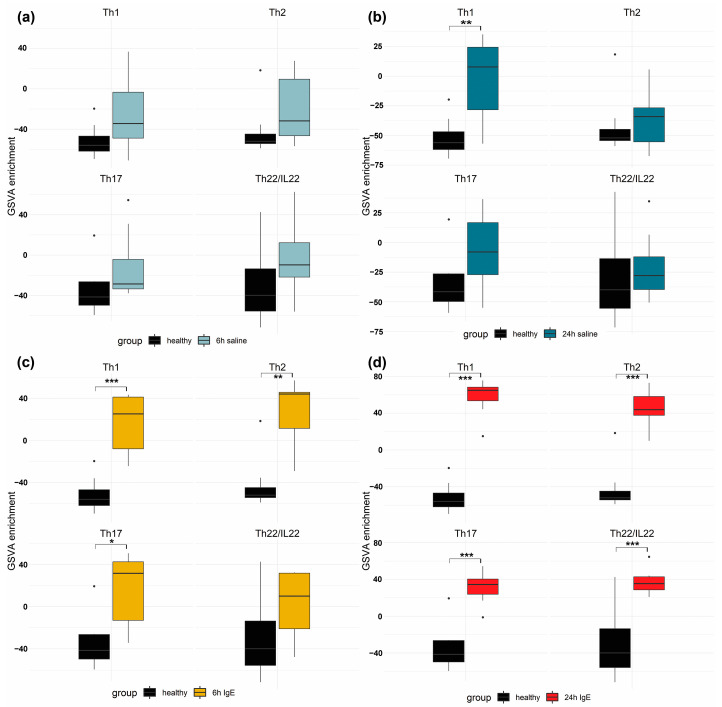
Gene Set Variation Analysis (GSVA) scores for T-helper (Th) 1, 2, 17 and 22/IL-22 immune pathway for each group (saline or anti-canine IgE) and time (6- or 24-h) versus healthy demonstrated in vertical box and whisker plots. * implies *p*-adj < 0.05, ** implies *p*-adj < 0.01, *** implies *p*-adj < 0.005. (**a**) 6-h saline vs. healthy with no significance. (**b**) 24-h saline vs. healthy with only Th1 showing significance. (**c**) 6-h anti-canine IgE vs. healthy with significance in Th1, Th2 and Th17 gene groups. (**d**) 24-h anti-canine IgE vs. healthy with significance in all GSVA gene groups.

**Figure 4 vetsci-11-00109-f004:**
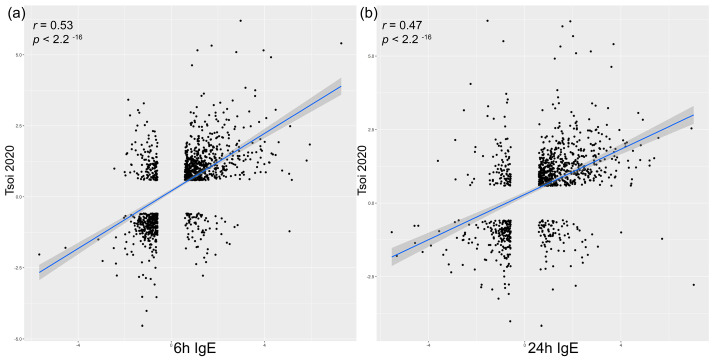
Spearman’s rank test of differentially expressed genes (DEGs; −1.5 ≥ FC ≥ 1.5, FDR < 0.05) between 6- (**a**) and 24-h (**b**) anti-canine-IgE reactions and acute skin lesions of spontaneous human atopic dermatitis [18]. Each dot represents a DEG shared between the spontaneous human atopic dermatitis study and 6- (**a**) and 24-h (**b**) anti-canine-IgE reactions. The line shown is a fitted loess curve corresponding to the high monotonic relationship of DEGs by Spearman’s rank testing.

## Data Availability

Data is contained within the articles and Appendix A. The raw sequencing data is available upon request.

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
