# Peer review of "Characterization of the Pro-Inflammatory and Pruritogenic Transcriptome in Skin Lesions of the Experimental Canine Atopic Acute IgE-Mediated Late Phase Reactions Model and Correlation to Acute Skin Lesions of Human Atopic Dermatitis"

_vetsci, 2024, doi:10.3390/vetsci11030109_

Round 1

Reviewer 1 Report

Comments and Suggestions for Authors

A highly valuable work with a lot of new and necessary information, precisely elaborated, necessary for scientists in the field of veterinary dermatology as well as comparative medicine

Author Response

Reviewer 1

A highly valuable work with a lot of new and necessary information, precisely elaborated, necessary for scientists in the field of veterinary dermatology as well as comparative medicine

Answer to the comment: Thank you

Reviewer 2 Report

Comments and Suggestions for Authors

Dear authors,

Congratulations for your work! The results are pretty impressive, with good impact in practice.

But I suggest you to add few more references, especially from the last 5 years!

Best regards!

Author Response

Reviewer 2

Dear authors,

Congratulations for your work! The results are pretty impressive, with good impact in practice.

But I suggest you to add few more references, especially from the last 5 years!

Best regards!

Answer to the comment: Thank you, we have added more references but if you feel we need other references, please specify which ones and we are happy to add these.

Reviewer 3 Report

Comments and Suggestions for Authors

Dear Authors,

Your paper is of high quality and clear exposed. You just have to control few points

lines 96, 121, 129, add USA

line 109 please add the formalin %

Author Response

Reviewer 3

Your paper is of high quality and clear exposed.

You just have to control few points

lines 96, 121, 129, add USA

Answer to the comment 1. Thank you, yes we have control points that need to be addressed at proof reading and we have added “USA”

line 109 please add the formalin %

Answer to the comment 2: Thank you, we have added “formalin”